# Enrichment of Cookies with Fruits and Their By-Products: Chemical Composition, Antioxidant Properties, and Sensory Changes

**DOI:** 10.3390/molecules28104005

**Published:** 2023-05-10

**Authors:** Anna Krajewska, Dariusz Dziki

**Affiliations:** Department of Thermal Technology and Food Process Engineering, University of Life Sciences in Lublin, 31 Głęboka St., 20-612 Lublin, Poland; anna.krajewska@up.lublin.pl

**Keywords:** biscuits, fruits, byproducts, chemical composition, bioactive compounds, antioxidant properties, consumer acceptability

## Abstract

Cookies made from wheat have become increasingly popular as a snack due to their various advantages, such as their convenience as a ready-to-eat and easily storable food item, wide availability in different types, and affordability. Especially in recent years, there has been a trend towards enriching food with fruit additives, which increase the health-promoting properties of the products. The aim of this study was to examine current trends in fortifying cookies with fruits and their byproducts, with a particular focus on the changes in chemical composition, antioxidant properties, and sensory attributes. As indicated by the results of studies, the incorporation of powdered fruits and fruit byproducts into cookies helps to increase their fiber and mineral content. Most importantly, it significantly enhances the nutraceutical potential of the products by adding phenolic compounds with high antioxidant capacity. Enriching shortbread cookies is a challenge for both researchers and producers because the type of fruit additive and level of substitution can diversely affect sensory attributes of cookies such as color, texture, flavor, and taste, which have an impact on consumer acceptability.

## 1. Introduction

Wheat cookies have gained widespread popularity as a snack food owing to their convenience as a ready-to-eat and easily storable product, wide availability in diverse varieties, and affordability. The ease of preparation, consumption, portability, and extended shelf life of cookies make them a popular choice among consumers of all ages [1,2]. Nevertheless, they are frequently composed solely of a blend of butter, sugar, and flour. On the one hand, this implies that they constitute a high-calorie food; on the other hand, they possess low levels of dietary fiber and bioactive compounds. Presently, a substantial proportion of consumers exhibit awareness regarding the link between inadequate nutrition and the emergence of diet-associated disorders. Consequently, they are seeking out functional foods that are rich in nutrients and have the potential to exert favorable effects on their physiologies [3]. Especially in recent years, there has been a growing interest in finding innovative and sustainable ways to use fruit byproducts (FBP) as a means of reducing waste and promoting circular economy practices. Fruits and FBP are rich sources of beneficial substances beyond their macronutrient (carbohydrates, proteins, and fats) and micronutrient (vitamins and minerals) contents. These substances consist of bioactive compounds such as carotenoids, sterols, phenolic compounds (including flavonoids and non-flavonoids), and dietary fiber [4]. Recent studies have provided valuable insights into the potential health-promoting effects of bioactive compounds present in fruits and FBP: anti-aging [5], antioxidant [6,7], anticancer [8,9], antiproliferative, anti-neurodegenerative, anti-inflammatory [10], antidiabetic [11,12], antimicrobial [9,13,14], anti-inflammatory [15,16], hepatoprotective [17,18], and neuroprotective [8,19] (Figure 1). These beneficial properties of fruits can make them valuable ingredients in the production of functional food products such as shortbread cookies. On the other hand, enriching shortbread cookies with powdered fruits and FBP usually requires drying and grinding them [20,21]. However, this enrichment process can cause changes in the cookies’ physical properties, including shape, color, and hardness [22,23]. It can also affect sensory perceptions such as taste, smell, and texture, potentially leading to a decrease in overall quality [21]. Therefore, it is crucial to determine the precise level of each additive while considering both health aspects and consumer acceptability.

The aim of this study was to examine the current trends in adding fruits and FBP, while focusing on the variations in antioxidant content, chemical composition, and sensory properties. In particular, scientific papers published within the past six years were mainly analyzed utilizing various databases, such as Scopus, Web of Sciences, and Google Scholar.

## 2. Changes in Chemical Composition and Antioxidant Activity

### 2.1. Cookies with Powdered Fruit Additives

The enrichment of shortbreads with fruits and FBP significantly changes the chemical composition and antioxidant capacity of cookies. These changes depend both on the type and amount of the additive used. In recent years, this topic has gained a lot of attention. As shortbread cookies require a small amount of water to produce, the addition of dried and powdered fruits is required (Figure 2) to enrich the cookies with bioactive compounds.

The antioxidant capacity of fruits is associated with the presence of antioxidants, including polyphenolic compounds, which may exhibit various mechanisms of action [24]. Hence, there is a need to analyze antioxidant activity using different methods [25]. Borczak et al. [21] studied the effect of a 5% addition of freeze-dried and powdered fruits of elderberry, wild rose, rowanberry, sea buckthorn, hawthorn, and chokeberry on the properties of cookies. Elderberries (*Sambucus nigra* L.) contain anthocyanins: cyanidin 3-sambubioside and cyanidin 3-glucoside, flavanols: rutin, isoquercetin, astragalin, phenolic acids: chlorogenic and sinapic acid, as well as organic acids [26]. The most potent antioxidants in rosehip (*Rosa canina* L.) are ascorbic acid and polyphenols, specifically cyanidin, catechin, quercetin, and gallic acid [27]. Rowan fruits (*Sorbus aucuparia* L.) contain procyanidin B1, carotenoids (including zeaxanthin and β-cryptoxanthin), catechin and epicatechin, ferulic acid methyl ester, and organic acids [28]. The antioxidant constituents of sea buckthorn (*Hippophae rhamnoides* L.) are flavonoids such as kaempferol, catechin, epicatechin, and isorhamnetin, phenolic acids including *p*-coumaric, gallic and caffeic acids, and tocopherols [29]. Hawthorn fruit (*Crataegus* L.) is high in hyperoside, chlorogenic acid, and quercetin [30]. Chokeberry (*Aronia melanocarpa* L.) contains anthocyanins (such as cyanidine 3-glucoside and 3-galactoside), flavonols (primarily quercetin glycosides), and phenolic acids (chlorogenic acid and neochlorogenic acid) [31]. It is therefore not surprising that the fortified cookies demonstrated significantly higher antioxidant activity (AA) than the control shortbreads, using the ABTS (2,2-azinobis(3-ethyl-beznothaizoline-6-sulfonic acid) and the ferric reducing ability of plasma (FRAP) methods. Cookies containing chokeberry showed the highest AA measured using ABTS, which was 15.22 µmol troloxg^−1^ dm (dry mass). Cookies with wild rose had the highest reducing potential in the FRAP assay (26.12 µmol Fe (II) g^−1^ dm). Fortification with each additive improved the total polyphenol content compared with the control. Cookies enriched with wild rose showed the highest concentration of the phenolic compounds (256.44 mg 100 g^−1^ dm). Aside from significant quantitative differences, there was also a difference in the type of phenolic compounds found in cookies with specific additives. For example, the presence of anthocyanins was only revealed in products enriched with chokeberry (idaein), elderberry, and hawthorn (kuromanine). The authors emphasized that the concentration of polyphenols in enriched cookies is lower than the amount of these compounds in the same amount of fruits themselves [21]. The effect of the applied temperature and exposure times could explain the decrease in the content of polyphenolic compounds [32]. Additionally, the stability of anthocyanins is related to their chemical structure [33], and their degradation could be affected by temperature and oxygen (through oxidative mechanisms and enhancing the reaction catalyzed by oxidative enzymes) [34]. Moreover, the content of carbohydrates, protein, fat, dietary fiber, and ash in individual fortified cookies tended to vary significantly. The highest amount of dietary fiber was found in sea buckthorn shortbreads (6.36% dm), while the lowest was in elderberry cookies (1.69% dm). The difference was likely due to the varying amounts of dietary fiber present in the different raw materials used [31,32,35,36]. Additionally, the degree of reduction of acrylamide, which is formed in carbohydrate-rich foods prepared at high temperatures, was measured. This compound is considered carcinogenic, neurotoxic, and mutagenic [37]. Importantly, the acrylamide content was reduced as a result of cookie enrichment [21]. Considering the conducted research, it can be concluded that enriching cookies with the analyzed additives not only contributes to improving their nutritional value and enhancing antioxidant properties but can also lead to a reduction in food-borne toxins such as acrylamide. In particular, the powder from chokeberry fruit showed the highest ability to reduce this compound. Importantly, cookies enriched with this fruit also showed the strongest antioxidant activity against ABTS free radicals. However, it is worth emphasizing that including some antioxidants in the food matrix can enhance the development of acrylamide [37].

Sady and Sielicka-Różyńska [38] assessed the quality of cookies after replacing 5, 10, and 15% of the dough weight with freeze-dried and powdered chokeberry (FPC). The enriched cookies contained less fat and more reducing sugars than the control. Furthermore, in cookies with chokeberry, the amount of ash increased, except for 5% of the fortifications. The concentration of protein in the analyzed products did not change significantly under the influence of the additive. Most importantly, the fortified products manifested a considerably higher amount of phenolic compounds and antioxidant activity (AA) determined by the DPPH (2,2-diphenyl-1-picrylhydrazyl) and FRAP methods compared to the control sample. According to the research, cookies containing 10% FPC had twice as much reducing power as those containing 5% (FRAP assay). The total content of phenolic compounds (TPC) and AA increased in direct proportion to the FPC content in the recipe. There was an increase in TPC from 14.1 mg GAE (gallic acid equivalent)/100 g dm in the control sample to 481.1 mg GAE/100 g dm in cookies containing 15% FPC. The significant increase in TPC indicates that powder from chokeberry fruit can be a very valuable and functional addition to food, especially shortbread cookies. Chokeberry fruit is especially rich in many bioactive compounds such as anthocyanins (up to 1% of dm), proanthocyanidins, flavonoids, and phenolic acids, represented mainly by chlorogenic and neochlorogenic acid [39]. This can explain the considerable increase in TPC in enriched cookies. Importantly, it has been proven that bioactive compounds in chokeberry fruit, such as phenolic acids and flavonols, are thermostable, whereas degradation of anthocyanins mainly occurs [40]. There are additional factors that support the use of aronia pulp for enriching cookies.

Other researchers [41] assessed the impact of adding freeze-dried and ground Japanese quince (*Chaenomeles japonica*) fruits on the properties of cookies. Japanese quince is a source of polyphenolic compounds, such as catechin, epicatechin, chlorogenic acid, procyanidin B1, and procyanidin B2, which, together with the biologically active form of vitamin C, determine the antioxidant potential of the fruit [42]. The 0.5% addition to the cookies resulted in a two-fold increase in antiradical activity against DPPH, while the 9.5% addition resulted in a 3.5-fold improvement over the control. Surprisingly, cookies enriched with 9% of the quince powder had a more powerful radical scavenging effect than the same amount of freeze-dried Japanese quince fruit. The authors explained this phenomenon by the enrichment of cookies with polyphenols, as well as the creation of Maillard reaction compounds. These reactions occur between amino acids and reducing sugars, resulting in the formation of brown pigments and aroma compounds, ultimately leading to an increase in the antioxidant activity of the enriched cookies [41]. However, during the 16 weeks of the cookie’s storage, there was a decrease in AA. It is important to note that the fortification reduced the loss of antioxidant activity in the storage cookies from 33% in the control sample to roughly 15% in the enriched cookies. The presence of procyanidins, which are assumed to improve antioxidant, microbiological, and thermal stability, could explain the observed result [43]. Moreover, the addition of Japanese quince increased the content of heptanal, hexanal, octanal, and 2-heptenal. With prominent notes of lemon and sour smells, acetic acid predominated in the volatile profile [41].

The presented paper reveals that the powder obtained from freeze-dried *Chaenomeles japonica* is a valuable additive to wheat cookies in terms of phenolic content and AA. Henceforth, upcoming research should contemplate the potential of enhancing cookies with quince powder acquired by various drying techniques. This is justified by the fact that Turkiewicz et al. [44] found that vacuum-drying of quince produces a final product with similar levels of phenolics and higher AA compared to the freeze-drying process of these fruits.

Zarroug et al. [45] replaced wheat flour with 15, 30, 45, and 100% pulverized African jujube fruit (*Zizyphus lotus*) in the production of cookies. The chemical composition of this fruit includes sugars (mainly fructose and glucose), volatile substances, and polyphenolic compounds (such as *p*-hydroxybenzoic acid and quercetin 3-*O*-rhamnoside-7-*O*-glucoside) [46]. The powdered additive improved the antioxidant properties (antiradical activity against DPPH and reducing power) of shortbreads [45]. The AA was measured as an inhibition concentration (IC_50_ or EC_50_) index, which indicates the concentration of the antioxidant needed to remove 50% of the DPPH free radicals [47]. Hence, the lower the IC_50_ or EC_50_ values, the higher the antioxidant activity [48,49]. In the case of radical scavenging ability, the IC_50_ ranged from 454 µg/mL (control cookies) to 95 µg/mL (100% cookies with 100% replacement of flour), while for reducing power this index varied from 950 µg/mL (control) to 320 µg/mL (100%) [45]. *Zizyphus lotus* fortification also improved the total contents of polyphenols, flavonoids, and condensed tannins, which depended on the additive amount. With the addition of powder, the fat and ash content improved, while the amount of protein decreased. It is worth mentioning that in this work, the authors attempted to produce cookies with a complete replacement of wheat flour with African jujube fruit powder. However, this resulted in an increase in the water content and water activity in the final production, which may have a negative impact on the shelf life of the cookies. It is also interesting to note that no significant effect of this additive on the caloric value of the cookies was observed, which can be explained by the much higher fat content in the powdered *Zizyphus lotus* fruit compared to wheat flour, as well as a high carbohydrate content. In another study, Pawde et al. [50] prepared cookies with 30, 40, 50, and 60% substitution of wheat flour by adding air-dried and powdered pitaya (*Hylocereus undatus*), also called dragon fruit. The plant comes from Mexico and Central America. Its fruits are a valuable source of soluble and insoluble fiber, sugar (glucose, fructose), organic acids (malonic acid and citric acid), phenolic acids (caffeic, ferulic and protocatechuic acids), amino acids, minerals (magnesium, potassium, calcium, copper, iron), and vitamins (C, K1, D2, biotin) [51,52]. The fiber concentration of the 60% powdered cookies was approximately six times higher compared to the un-enriched cookies. That difference was related to the crude fiber content of pitaya powder (15.7%). The authors noted that a daily intake of 100 g of fortified cookies could fulfill 20% of the RDI (Recommended Daily Intake) for fiber. Introducing this ingredient into the diet is essential because of its association with reducing the risk of cardiovascular disease [53]. Furthermore, the fortification enhanced the ash content of the cookies proportionally to the fruit concentration. It also resulted in a decrease in protein levels. Other authors [54] prepared cookies with papaya powder (*Carica papaya*). The papaya fruit is a good source of vitamins A, C, and E, minerals (magnesium, potassium, pantothenic acid, and folic acid), fiber, papain, and enzymes that improve digestive tract function [55]. The AA and TPC of cookies increased with the percentage of the additive. Additionally, the high DPPH radical inhibition was potentially correlated with the TPC (1.705 µg GAE mg dm) and total flavonoid content (1.017 µg CE (catechin equivalent)/mg dm) of the papaya pulp product [54]. Moreover, protein and fat concentrations increased after the supplementation. The dietary fiber content of the fortified cookies was 3.35%. Enrichment also improved the concentrations of sodium, iron, and vitamin A in the final product. It is worth emphasizing that in both cited works [50,54], the additives used, besides enhancing antioxidant properties, resulted in an increase in the fiber content of the enriched cookies. This effect was particularly noticeable in the case of enriching cookies with dragon fruit powder. Importantly, the use of this additive as a substitute for wheat flour, even up to 50% of the total amount, did not have a negative impact on the quality of the cookies. This indicates that it can be a very valuable addition from a nutritional standpoint for fortifying such products.

Other researchers [56] replaced sugar with date powder in ratios of 90:10, 80:20, 70:30, 60:40, and 50:50 (whole wheat flour: date palm fruit powder) in cookie making. Dates are a valuable source of carbohydrates, amino acids, fiber, and phenolic compounds such as ferulic acid, hydrobenzoic acid, luteolin, and quercetin [57]. These fruits are also rich in calcium, copper, magnesium, potassium, iron, and manganese [58]. The incorporation of date powder increased the amount of crude fiber in the cookies, except for the 10% enrichment level. Its content depended on the amount of the mixture used and ranged from 1.72% (10% powder) to 2.39% (50% powder). However, there was no significant difference in crude fiber concentration between cookies containing 10% of the admixture and between whole wheat flour alone and refined flour (control) products. This is likely due to the high standard deviation values obtained by the authors during the determination of fiber content, which subsequently affected the results of the analysis of variance. The date powder also increased the ash content (from 1.63% in a sample containing only whole wheat flour to 3.07% in cookies with 50% date powder), demonstrating a significant mineral content of the product. The fat content also increased under the influence of this additive [56]. Date fruits have high energy values due to the presence of unsaturated fatty acids such as oleic acid, palmitoleic acid, linoleic acid, and linolenic acid [59]. The control had a higher carbohydrate content (53.43%) than the fortified ones (except for the cookies with 50% powder), probably due to the use of sugar in the formulation of control cookies. Nonetheless, the percentage of carbohydrates in the chemical composition of the fortified products ranged from 52.12% (10 and 20% powder) to 55.31% (50% powder). The protein content was also notably reduced by the addition of dates [56].

To summarize this chapter, it can be concluded that the fruit powders described are a valuable substitute for wheat flour from a nutritional standpoint, as they increase the nutritional and health value of shortbread cookies. However, it is worth noting that the production process for these enriched cookies often requires the addition of powdered and crushed fruits, which significantly increases their production costs. Therefore, it may be more advantageous to use by-products from the fruit processing industry, which are not only a cheaper raw material but also a valuable source of many bioactive compounds and fiber. Additionally, they are often pre-dehydrated. This topic will be further discussed in the next chapter of the manuscript.

### 2.2. Cookies with the Addition of Fruit By-Products 

In recent years, several studies have investigated the possibility of using FBP as nutritional additives in shortbread cookies. Some of these studies specifically focused on apple by-products, which are a common source of phenolic compounds, including quercetin and its glycosides, epicatechin, procyanidins, anthocyanidins, and phlorizin [25]. Zlatanović et al. [60] prepared finely ground pomace flour (weight share was 27.78% for particles sized between 160 µm and 300 µm) and coarsely ground pomace flour (weight share was 28.32% in the range of 500 µm to 1 mm). They then analyzed the effects of replacing wheat flour with 25%, 50%, and 75% pomace on cookie production immediately after preparation and after one year of storage. The contents of dietary fiber, phenolic compounds, and flavonoids in the products were proportional to the content of the admixture used. The amount of dietary fibers in functional products with the addition of 25%, 50%, and 75% pomace flour was 10.1%, 20.6%, and 31.4% dm, respectively, while the control product contained only 1.7% dm. On this basis, the manufactured cookies met the criteria for high-fiber products. This was expected since nearly 40% of the dry weight of apple pomace is fiber [61]. Antiradical activity against DPPH increased 3.2-, 4.0-, and 5.5-fold (coarse) and 2.9-, 4.0-, and 4.5-fold (fine) for 25, 50, and 75% enriched cookies, respectively, compared with the control. In contrast, the scavenging properties were 4.4, 7.4, and 8.9 times (coarse) and 4.6, 6.7, and 8.5 times (fine) stronger against the ABTS radical, respectively. Interestingly, the products enriched with coarse flour contained more dietary fiber, polyphenols, and flavonoids than the cookies fortified with fine apple pomace flour. They also performed better in tests of antioxidant activity. The greater stability of the active ingredients (even after one year of storage) in the larger particles could be due to reduced contact of the polyphenolic substances with oxygen and light in the food matrix. Moreover, dietary fiber content did not change significantly after 12 months of storage, while the decreases in the contents of polyphenols, flavonoids, and AA were significant. Similarly, the results showed that the functional products retained their properties better than the control cookies. Nakov et al. [62] replaced 4%, 8%, 16%, 24%, and 32% of wheat flour with dried and ground apple peel in the production of cookies. The products enriched with apple peel showed stronger AA (DPPH and FRAP), and higher ash (by 21%) and fat content compared to the control. The cookies without additives contained 146.15 µg GAE/g dm of phenolic compounds. The incorporation of a small amount of apple peel (4%) resulted in a significant increase to 250.6 µg GAE/g dm. Products with 32% additives contained 622.12 µg GAE/g TPC. Total fiber content increased steadily from 2.2% dm (control) to 10.4% dm (32% additive). The soluble fraction grew more than the insoluble fraction. Soluble fibers, including hemicelluloses and pectins, have various fermenting abilities, and many of them induce the proliferation of health-promoting microflora. Insoluble fiber, including cellulose, lignin, and resistant starch, is highly insoluble in water and not digested by digestive enzymes, but fermented to varying degrees by intestinal bacteria [63]. The structure of the polysaccharide affects its solubility [64]. The protein concentration in the cookies with 32% admixture was reduced by 58% compared to the control. Proteins coagulate during baking and improve the structure of gluten, so their quality and content are one of the determinants of the quality of the cookies [62]. Presented papers [60,62] showed that both apple pomace and apple peel are valuable additives to wheat cookies and significantly increase the TPC, fiber content, and AA of the enriched product. Importantly, it was also proven that the use of coarse particles of apple by-products has a positive influence on the stability of bioactive ingredients during cookie storage. This indicates that the particle size of the used admixture should be taken into consideration for future studies.

Tarasevičienė et al. [65] replaced wheat flour with 10, 15, and 20% raspberry, currant, and strawberry powdered pomace in cookies. Raspberry fruit contains significant amounts of vitamins A and B, ellagitannins, anthocyanins, and iron [66]. Red currants are a natural source of vitamins (ascorbic acid), trace elements, organic acids, and polyphenolic compounds such as syringic acid and catechin [67,68]. Strawberry fruits are rich sources of anthocyanins (elargonidin-3-glucoside), flavanols (kaempferol, quercetin), and flavonols [69]. The enriched products exhibited significantly higher levels of dietary fiber and its fractions compared to the control. The highest total dietary fiber content was found in cookies with 20% raspberry pomace (15.66% dm). These products contained 17.30% dm neutral detergent fiber (cellulose, hemicellulose, and lignin), 16.15% dm acid detergent fiber (cellulose and lignin), and 18.80% dm modified acid detergent solution. Furthermore, the fortification increased the ash concentration while decreasing the protein content. The cookies containing 20% red-currant pomace had the highest ash content of 3.85% dm. Moreover, water-soluble carbohydrates were the primary component of the analyzed products, with values ranging from 24.30% dm (control) to 29.25% dm (20% strawberry pomace cookies). Tagliani et al. [70] reviewed the optimal conditions for cookie production using dried and ground blueberry pomace. The by-product came from industrial juice production. Blueberries are abundant in anthocyanins, phenolic acids (for example hydroxycinnamic acids), flavonols, dietary fiber, vitamins (C, B-complex, E, A), B-carotene, lutein, zeaxanthin, and minerals (zinc, iron, selenium, magnesium) [71]. It has been found that blueberry pomace possesses robust antioxidant characteristics, primarily attributed to the presence of polyphenols and anthocyanins. The high fiber content present in blueberry pomace is also responsible for its high levels of polyphenolic compounds and antioxidant properties. The hydrolyzing fraction influenced the antioxidant capacity, which is conditioned by the biocomponents bound with fiber. In addition, increasing baking temperature and dough thickness decreased antioxidant activity (tested with ABTS) and total polyphenolic compound content. Cookies with 9% dietary fiber baked at 180 °C with a dough thickness of 0.50 cm and those baked at 170 °C with a dough thickness of 0.75 cm showed the highest AA and polyphenol content [70]. Based on the papers cited in this paragraph, it can be concluded that the pomace from the mentioned berries is a valuable source of additives in shortbread cookies, which can significantly enrich the final products with antioxidant fiber. Moreover, the presented papers offer a great approach to reassessing the waste produced by the juice industry by creating a novel component that could potentially be utilized as an ingredient in functional cookies. However, it is also essential to establish the optimal baking conditions since factors such as process temperature or dough thickness significantly influence the AA of the final product. Rani et al. [72] developed functional cookies with orange peel powder (5, 10, 15, and 20%). Orange peels are a valuable source of vitamin C, dietary fiber, essential oils, phenolic acids (such as gallic acid and protocatechuic acid), flavonoids (including catechins and epigallocatechins), and organic acids [73]. Fortified cookies contained significantly more total dietary fiber (from 8.33 to 13.33% dm), insoluble dietary fiber (from 5.43 to 7.36% dm), and soluble dietary fiber (from 2.82 to 6.00% dm) compared with the control shortbreads. Additionally, the cookies with the FBP also showed stronger antioxidant properties (DPPH method) and higher TPC. The ash content increased, while the levels of protein and fat decreased under the influence of incorporation. The authors made a recommendation of popularising cookies enriched with orange peel due to the high concentrations of fiber, minerals, and antioxidants. Imeneo et al. [74] analyzed the effect of fortifying cookies with fresh lemon peel and freeze-dried lemon pomace extract produced during the manufacturing of juices and essential oils. Lemon peels contain significant amounts of polyphenolic compounds, especially flavanones, such as eriocitrin and hesperidin [75]. Lemon peels are also rich in phenolic acids (such as ferulic acid and vanillic acid), flavanones (eriocitrin), flavonones (luteolin), flavonols (quercetin 3-*O*-rhamnoside), flavanones (such as hesperidin), carotenoids, and volatiles [76]. The fortified cookies showed greater antioxidant potential (as evidenced by DPPH and ABTS assays) and higher polyphenol content than the control samples. Fortification resulted in a doubling of the TPC (from 15 mg GAE 100 g^−1^ dm in the control sample to about 30 mg GAE g^−1^ dm in the supplemented cookies). Cookies with additives exhibited a longer induction time, implying a higher resistance to lipid oxidation. This ability affects the longer preservation of food. The antioxidant effect of lemon is related to the prevention of the lipid component of the matrix during the oxidation reaction [77] and presumably to the high amount of ascorbic acid [78]. Laganà et al. [79] studied the effect of replacing wheat flour in manufactured cookies with pressed and dried bergamot pulp. Bergamot is a source of polyphenolic compounds, such as neoeriocitrin, rutin, neohesperidin, and naringin [80]. A citrus by-product known as *Pastazzo* caused significant disposal problems in recent years [81]. The researchers indicated that as the percentage of bergamot pulp increased, the total phenolic compound content increased (from 0.14 mg GAE g^−1^ dm for the control to 3.64 mg GAE g^−1^ dm for the 15% addition). The percentage of polyphenol content correlated with the amount of additive. Flavonoids showed a similar trend. Hesperidin, naringin, and neoeriocitrin were present in the highest amount of *Pastazzo*. The fortified cookies also demonstrated more intense antiradical activity in the ABTS and DPPH assays compared to the control. The products enriched with 10 to 15% pulp reveal a non-linear increase in scavenging ABTS radicals. The DPPH test manifested an unclear relationship between the percentage of additives and the antioxidant properties of the cookies. The cookies with 5% additive showed higher activity against DPPH (1318.5 mg Trolox equivalent (TE) 100 g^−1^) than those with 15% additive (1244.41 mg TE 100 g^−1^). The authors pointed out the recommendation to use at least two methods to measure the antioxidant properties of citrus, because of the possible synergistic effects of antioxidants. They also emphasized that the tests for measuring antioxidant potential use different molecules as reagents, which may lead to different results.

Summarizing the results concerning the possibility of enriching cookies with citrus, it is clear that orange and lemon peels, as well as lemon pomace and dried bergamot pulp, are valuable additives, especially in terms of increasing the TPC and AA of shortbreads. However, bergamot powder is especially recommended as the richest source of biologically active compounds. Importantly, it was also proven that fresh lemon peel has a similar positive effect on the cookies’ phenolic content and AA as dried and powdered orange peel (at the same percentage of wheat flour replacement as fresh lemon peel).

Other researchers [82] observed the effect of enriching cookies with banana peel flour (through the substitution of 7.5, 10, 12.5, and 15% of wheat flour) on the nutritional and antioxidant properties. Banana peels comprise approximately 30–40% of the weight of the fruit. It is a valuable source of carbohydrates, fiber, protein, potassium, amino acids, fatty acids, and polyphenolic compounds, for example, gallocatechin [83,84]. The fortified cookies contained more TPC and demonstrated stronger AA against DPPH, as well as higher inhibition of lipid peroxidation than the control products. The TPC content ranged from 0.282 (control) to 0.921 mg GAE/g dm (cookies enriched with 15% banana flour). The reducing power against DPPH increased with the increase in pomace concentration, from 62.02% (7.5% of added) to 70.29% (15% of added). The inhibition of lipid peroxidation showed a similar tendency, which ranged from 51.51% (7.5% addition) to 71.84 (15% addition) for the fortified cookies. The authors observed a significant decrease in the antioxidant activity of the cookies during three months of storage, which could be due to the degradation of phenolic compounds and carotenoids. In addition, the moisture and ash content in the enriched cookies improved, while the protein and fat content decreased. It is worth mentioning that the authors of the cited paper [73] did not provide information about the particle size of the banana peel flour used. As previously mentioned [52], the particle size of the fruit admixture used can affect both the physicochemical properties of cookies and their stability during storage. Ning et al. [29] added flour from the passion fruit peel to cookies. Passion fruit contains significant amounts of dietary fiber, proanthocyanidins, carotenoids, and flavonoids such as apigenin and luteolin [85]. The fruit flour supplementation increased the contents of polyphenols and dietary fiber in the cookies, and their concentration depended on the percentage of admixture used. The TPC content was 2.3 mg FAE (ferulic acid equivalent)/g dm of the control and ranged from 2.6 (for the cookies with 3% addition) to 2.9 mg FAE/g dm (9%). The degree of DPPH reduction ranged from 21.5% (control) to 46.8% (9% addition). The fiber content increased as well. The authors found that phenol-fiber compounds have higher antioxidant potential compared to phenols or fibers alone and play a crucial role in protecting the cell wall by creating physicochemical barriers. In other studies, Garcia et al. [86] substituted 10, 20, and 30% whole wheat flour with the addition of passion fruit peel flour in the production of cookies. The enhanced product contained more crude fiber (2.76–3.51% dm) and ash (2.88–3.20% dm) than the control (1.72% and 1.74% dm, respectively). In addition, the fortification contributed to a decrease in the total carbohydrate, lipid, and caloric value of the cookies. The cited works indicate that the use of passion FBP has a positive effect on the nutritional value of cookies. Additionally, it increases the resistant starch content in cookies and consequently reduces the starch digestion rate. Toledo et al. [87] included dehydrated and milled pineapple center axis, apple endocarp, and melon peel at concentrations of 5, 10, and 15% in cookies. Pineapple (*Ananas comosus*) is rich in dietary fiber, sugars, vitamins (ascorbic acid, niacin, thiamine), minerals (e.g., potassium, magnesium, copper, manganese), organic acids, volatile ingredients, and bromelain, a proteolytic enzyme that improves digestion [88]. Apple endocarp has a high nutraceutical value. Apples contain sugars (with the highest amount being fructose, followed by glucose, and the least amount being sucrose), organic acids, vitamins (mainly vitamin C), minerals, and dietary fiber (about 2–3%, of which over 50% is pectin) [89]. Melon peels contain a significant amount of fiber, protein, ash, polyphenols (including luteolin 7-glycoside and 4-hydroxybenzoic acid, luteolin-6-*C*-glycoside and ferulic acid), and carotenoids (such as β-carotene and β-cryptoxanthin) [90]. The fruit admixtures significantly increased the ash content (except for no difference between the control and cookies containing 10% pineapple). The increase in ash concentration after the incorporation of additives was particularly noticeable with regards to melon peel, with a significant difference between the use of 5, 10, and 15%. Carbohydrate content was the highest of all macronutrients. It was expected due to the use of wheat flour as the main ingredient. Furthermore, cookies containing 5% pineapple had the most carbohydrates (69.41%), while those containing 15% melon had the least (63.54%). The authors justified the obtained results with the difference in the content of carbohydrates in pineapple (about 13%) and melon (6.5–9%). The second macronutrient, in terms of quantity, turned out to be lipids, which are related to the notable use of butter in the production of bakery products. The protein content ranged from 8.13 to 8.94%, with the cookies with pineapple as the central axis having the lowest amount. Moreover, there was a significant and highest increase in the value of fiber in cookies containing melon peel powder (4.67–6.46%) compared to other additives. The soluble fiber content in the cookies was higher than the insoluble fiber content. It is worth highlighting that the use of FBP enhanced the bioavailability of minerals and reduced the content of antinutrients in cookies. Particularly, the study demonstrated that melon by-product promoted cookies with higher levels of calcium, iron, and zinc. Unfortunately, few papers on cookie enrichment are devoted to these topics.

Other authors [22] made cookies using flour derived from the peel of the *Parinari curatellifolia* fruit. This plant is a valuable source of biocomponents, including flavonoids, saponins, steroid tannins, cardiac glycosides, and terpenes [91]. The peel flour had a positive effect on the total polyphenol and total flavonoid contents of cookies. The high concentration of these substances in cookies may be additionally related to the baking temperature, which contributes to the hydrolysis of complex phenols [92]. The total flavonoid content (TFC) increased proportionally to the amount of admixture used, from 0.028 mg CE (catechin equivalent)/g dm (control cookies) to 0.104 mg CE/g dm (20% additive). The chemical composition of *Parinari curatellifolia* had an expected effect on the flavonoid amount in the final products. It is worth mentioning that thermal treatment could be demonstrated to increase the level of biocomponents concentrated in the cell wall by expanding their release. The incorporation of peel flour also resulted in a significant improvement in the antioxidant potential of the cookies. The percentage of DPPH inhibition increased from 48.70% (control) to 94.72% (20% enriched cookies). The FRAP results ranged from 108.33 mg GAE/g dm (reference sample) to 162.67 mg GAE/g dm (15 and 20% additive). The authors associated the cookies’ strong antioxidant potential with high TPC and TFC. Moreover, under the influence of *Parinari curatellifolia* peel in fortified cookies, an increase in ash and crude fiber was detected, as well as a decrease in fat, crude protein, carbohydrates, and energy (kcal) [22]. To summarize, this paper demonstrates that the peel of *Parinari curatellifolia* is a valuable additive for fortifying cookies from a nutritional standpoint. However, the authors did not perform a quality evaluation of the cookies in the present study.

Grape pomace is the main by-product of winemaking and accounts for 15–20% of the total weight of the fruit. Considering that the cultivation of grapes (*Vitis vinifera*) is one of the most developed fruit crops in the world, the use of post-production waste is crucial from an industrial point of view [93]. Grape marc is used in the production of alcohol, fodder, and fertilizers, or it is recycled. It transpires, however, that most animals are unable to digest them, their use as fertilizer is uneconomical, and their disposal causes problems related to environmental pollution [94]. Grape pomace is a source of fiber (about 50% of dry matter), protein, minerals (mainly iron), and phenolic compounds such as anthocyanins (delphinidin-3-*O*-glucoside, petunidin-3-*O*-glucoside, and cyanidin-3-*O*-glucoside), flavonols (quercetin, kaempferol), phenolic acids, and catechins [95]. Hence, the use of grape pomace for food fortification is a valuable idea that has been the subject of numerous studies in recent years. As an illustration, Theagarajan et al. [96] developed functional cookies including 2, 4, 6, and 8% of dried and pulverized grape pomace. The addition of powder to the cookies improved the protein and ash levels. Their values vary according to the amount of admixture employed. However, the supplementation had no significant effect on the carbohydrates or calorific value of the cookies. The fat content of the enriched cookies differed significantly between the control and powder levels of 4, 6, and 8%. Similarly, there was no significant difference in fiber content between the 2% pomace cookies and the control. However, cookies with 4, 6, and 8% of the additive showed considerably more fiber than those with none. Furthermore, 6% pomace powder-containing products had a higher TPC (4.03 mg GAE g^−1^ dm) than products containing 4% admixture (3.42 mg GAE g^−1^ dm) and the control sample (3.41 mg GAE g^−1^ dm). In the DPPH test, enriched cookies demonstrated moderate anti-radical activity. For 6% powder products, the IC_50_ value was 0.72 mg/mL of extract, whereas it was 0.85 mg/mL of extract for 4% additive cookies [96]. The total anthocyanin content also depended on the concentration of the additive used. The total anthocyanin content was 0.85 mg/100 g dm in 6% pomace cookies and 0.72 mg/100 g dm in 4% pomace cookies. In addition, Fontana et al. [97] improved shortbread cookies with a mixture of pomace flour from three varieties of white grapes (Sauvignon Blanc, Chardonnay, and Gewürztraminer) in equal proportions. Enriched cookies were a source of fiber (6.05%), ash (1.59%), lipids (19.55%), and proteins (6.64%). The fortified cookies also had a high carbohydrate content (58.77%). It Is not surprising given that earlier studies have revealed a significant presence of glucose and fructose in grape marc [98]. Furthermore, the AA of the cookies was lower than that of grape flour. The difference was associated with the dilution of the pomace with other ingredients, antioxidant thermal sensitivity, and the effect of particle size on antioxidant potential. The AA of pomace-enriched products was 2.22 μmol TEAC.g^−1^ (DPPH assay) and 6.73 μMol Trolox.g^−1^ (ABTS assay) [97]. Other authors [23] used dried and ground grape peels and seeds to supplement cookies, replacing wheat flour with 5, 10, and 15% of the additive. An increase in the total content of dietary fiber in fortified cookies was proven, which correlated with the level of the admixture used. Dietary fiber concentrations in products containing peel components ranged from 5.72% (a 5% supplement) to 8.64% (15%). With seeds, fiber content varied from 6.60% dm (a 5% supplement) to 9.76% dm (15%), while the control sample (without the additive) resulted in 2.37% dm. The high fiber concentration of the cookies is due to the considerable total fiber content, which was 56.13% dm for peels and 58.80% dm for seeds. The presented works clearly indicate that grape FBP are valuable functional additives for enriching the nutritional and health value of cookies. Importantly, they are widely available by-products with high fiber contents exceeding 50%. It is also worth emphasizing that grape marc shows anti-microbial properties [99,100]. Therefore, future studies on their use for enriching cookies should also focus on this topic.

Asadi et al. [101] studied the properties of cookies enriched with dried and powdered chiku (*Manilkara zapota* L.) pomace. Chiku fruits perform nutritional functions due to their high content of proteins, carbohydrates, fats, fiber, and vitamins. They are a valuable source of minerals such as calcium, copper, potassium, iron, and zinc. In addition, chiku exhibits potent antioxidant properties due to the presence of phytochemicals in the edible parts of the plant. The most crucial *Manilkara zapota* components are β-carotene, violaxanthin, lutein, gallic acid, catechin, quercetin, and kaempferol [102]. The inclusion of chiku pomace led to a significant improvement in the AA of the cookies when tested with DPPH. The percentage of radical scavenging activity depended on the concentration of the admixture used. The radical inhibition of the enriched cookies ranged from 5.94% (4.5% of the additive) to 7.13% (12% of the additive), while the control value was 0.27%. There was also an increase in crude fiber, total dietary fiber (and its soluble and insoluble fractions), total protein, total fat, total energy, and total phenolic compound content. The TPC did not differ significantly in the cookies with 4.5, 7, and 9.5% of admixture. Products enriched with 12% pomace showed the highest TPC (0.97 mg GAE/g dm).

Considering the above, it should be noted that *Manilkara zapota* pomace can be a valuable functional additive to cookies. However, despite enriching the cookies with many bioactive compounds, it also increases the calorie content of the products. Additionally, these tropical fruits are found mainly in India, Mexico, and Central and South America [103]. Hence, it appears that using chiku pomace to enrich cookies globally is not a reasonable choice. Goji berries (*Lycium barbarum* L.) can be a valuable additive to the fortification of cereal products. The chemical composition of goji berries includes polysaccharides, vitamins, minerals, carotenoids (such as zeaxanthin, β-carotene derivatives, and lutein isomers), organic acids, and phenolic compounds. These compounds condition the health-promoting properties of the fruits. Goji berries are rich in flavonoids, such as quercetin-3-*O*-rutinoside and kaempferol-3-*O*-rutinoside, and phenolic acids, such as chlorogenic, *p*-coumaric, and caffeic acids [104,105]. Bora et al. [106] incorporated goji berry pomace powder into cookies and muffins because of the variety of bio-components in the raw material and the potential economic benefit of eliminating waste disposal costs. The researchers substituted wheat flour with 10, 20, 30, and 40% of pomace. The substitution improved the content of soluble and insoluble fiber and protein fractions, which confirms the possibility of using the proposed powder to obtain bakery products with increased nutraceutical potential. The soluble fiber content of the cookies has been increased from 1.9% (control) to 3.7% (40% supplementation), while the insoluble fiber content has been increased from 3.7% (control) to 11.4% (40% supplementation) and increased with increasing powder concentration. Fortification also significantly improved the ash content, which justifies the effect of the additive in enriching the final product with minerals. Importantly, a significant intensification of the content of free phenolic components was also observed under the influence of the inclusion of goji berry pomace. The content of free phenolic compounds in enriched cookies ranged from 83.5 µg/g dm (10% supplementation) to 323.0 µg/g dm (40% supplementation), while in the unadulterated sample, the result was 2.0 µg/g. The number of free phenolic compounds depended on the pomace powder content. Consequently, it can be concluded that goji berry pomace powder significantly enriches cookies with insoluble and soluble dietary fiber, minerals, and free phenolic compounds. It is worth highlighting that replacing 10% of wheat flour with powdered Goji berry pomace results in about a twenty-fold increase in free phenolic compounds in cookies.

Kiwi fruit contains dietary fiber, vitamins A, C, and E, folic acid, minerals, carotenoids (zeaxanthin, lutein), actinidain (a proteolytic enzyme that improves protein digestion), and polyphenolic compounds (caffeic, gallic, and syringic acids) [107]. Kiwi peel is also a rich source of chlorogenic, caffeic, ferulic, and vanillic acids, as well as catechins, quercetin, and rutin [108]. Wang et al. [108] discovered that kiwi peel powder could contain more TPC and total flavonoid content, and also has higher antioxidant activity than pulp powder. Mohamed [109] came to similar conclusions when testing the effect of substituting wheat flour (5, 10, 15, and 20%) with powder from the edible part of the kiwi and kiwi peels on the properties of cookies. The kiwi peel products contained more dietary fiber and ash than the control and cookies with the edible portion of the kiwi. The quantity of these components positively correlated with the level of substitution. The amount of crude protein in the cookies, in contrast, declined as the level of substitution increased in comparison to the control. There was also a slight reduction in carbohydrates after the fortification. The TPC, flavonoid, and tannin concentrations were improved as the powder percentage increased. The highest amounts of these compounds were determined in cookies fortified with 20% kiwi peel, which accounted for 166.08 mg 100 g^−1^ dm, 26.17 mg 100 g^−1^ dm, and 77.76 mg 100 g^−1^ dm, respectively. Furthermore, the antioxidant properties of the fortified cookies were higher than the control (DPPH and ABTS assays). Antiradical potential increased with the percentage of the supplement added. According to the studies mentioned above, the antioxidant power of fruit raw material contributes to the final product’s antiradical effect. In summary, we can conclude that both the powdered edible parts of the kiwi and the crushed peel of this fruit can be valuable functional additives to shortbread cookies. Given that the powder derived from the peel has a higher concentration of bioactive compounds when compared to the flesh of the fruit, it appears to be a superior choice for enhancing the nutritional value of cookies. Additionally, it is a by-product of kiwi processing, thus a cost-effective functional supplement, although it requires processing into powder form.

Other authors [110] produced cookies composed of dried, powdered, and sieved cactus pear peel instead of 2.5, 5.0, 7.5, and 10% wheat flour. The cactus pear peel can represent from 48 to 52% of the total weight of the fruit [111]. As an agro-industrial by-product, it causes problems related to disposal and ecology [112]. This production waste is a valuable source of dietary fiber, vitamin C, and betalains [113]. The cookies with 7.5% admixture contained significantly more fiber (14.2%) than those without the addition (3.08%). Moreover, the fortification advanced the reduction of the DPPH radical in enriched cookies compared to the control [110]. It is related to the richness of the polyphenolic components of the cactus pear peel powder. Other researchers [20] enriched cookies with a by-product of processing camu-camu (*Myrciaria dubia*). They replaced wheat flour with 5, 10, 15, and 20% fruit waste. This exotic, Amazonian fruit is rich in vitamin C, carotenoids, and polyphenolic compounds such as anthocyanins, rutin, ferulic acid, and *p*-coumaric acid [4,114]. Camu-camu is typically consumed after being processed, which generates significant amounts of waste by-products. Chagas et al. [20] dried camu-camu co-product powder (including seeds, peels, and pulp residues) at 50, 60, and 70 °C in a forced-air oven. The powder increased the AA of cookies as measured by the FRAP and ORAC (oxygen radical absorbance capacity) methods. On the other hand, there were no significant differences in DPPH tests. The variations in the outcomes of diverse antioxidant assays can be ascribed to the fact that these assays assess distinct facets of the antioxidant capacity of the compounds under scrutiny. The DPPH assay quantifies the scavenging activity of stable free radicals, while the FRAP method illustrates the antioxidant potential of reducing iron ions, and the ORAC assay, being a competitive technique, measures the oxygen radical absorption capacity [115]. Cookies with a 20% addition of by-products demonstrated the highest AA in the FRAP test, which was 1.59 μmol trolox equivalent/g of powder. In the ORAC method, products containing 20, 15, and 10% camu-camu achieved the best activity. The results did not differ significantly from each other and amounted to 24.78, 21.14, and 25.38 μmol trolox equivalent/g of powder, respectively. The concentration of phenolic compounds depended on the content of the additive and ranged from 1.2 mg GAE/g dm (5% powder) to 4.4 mg GAE/g dm (20% powder) in enriched cookies. The result of the control was 0.7 mg GAE/g dm. Furthermore, the camu-camu by-product that was dried at a temperature of 70 °C exhibited a greater TPC in comparison to the by-products that were dried at lower temperatures. The observed outcome can be attributed to the potential cleavage of lignin-phenolic acid bonds or the degradation of lignin itself at higher drying temperatures. This may lead to the increased release of phenolic compounds from plant tissues. Additionally, elevated temperatures may inactivate enzymes, thereby increasing the recovery of phenolic compounds [116]. It indicates that the conditions of FBP (fruit by-product) preparation can also influence the properties of enriched cookies. However, only a few studies have addressed this issue.

To sum up this chapter, it should be noted that the incorporation of fruits, particularly FBP, into cookies helps to increase their fiber and mineral content. Furthermore, it considerably enhances the nutraceutical potential of the products by adding phenolic compounds with high antioxidant activity. Consequently, the enriched cookies can help prevent and reduce the risk of many diseases. In addition, incorporating FBP could offer significant economic benefits to fruit juice/concentrate producers who currently incur considerable expenses in disposing of FBP.

## 3. Sensory Properties

### 3.1. Cookies Enriched with Powdered Fruits

Consumer testing is crucial for the development of a product and proving that it has met presumed acceptance standards [117]. Sensory evaluation is a popular tool that could indicate consumers’ possible reactions to proffered cookies. In fact, the functionality of a product is not always a sufficient reason to buy shortbread cookies if the sensory quality is not satisfactory. In recently published papers, parameters such as taste, texture, aroma, aftertaste, and important overall acceptability are most often considered [45,106]. A nine-point hedonic scale is mainly used for consumers’ acceptance evaluation of different food products [38]. The quality, intensity, or type of sensory attributes are usually examined organoleptically by panels of trained testers on five- and ten-point scales [21,38]. Purchase intention for enriched cookies could be determined using a seven-point structured scale [117]. The physical and chemical characteristics of shortbread are related to sensory quality and consumer acceptability. Changes in these factors were demonstrated by the variability in sensory evaluations in recent years’ papers.

In a study by Borczak et al. [21], a trained group of panelists organoleptically evaluated (on a five-point scale) shortbread cookies enriched with fruit powders from elderberry, rosehip, chokeberry, sea buckthorn, hawthorn, and rowan. The following factors were considered quality parameters: taste (0.25), shape (0.2), aroma (0.2), surface (0.15), color (0.1), and consistency (0.1). It was found that the baked products enriched with rosehips received the total best score (4.40 points) when compared to the control sample (4.14 points). The cookies with the rowan additive received a high rating (4.19 points), which was not significantly different from those without the supplement. Although the other fortified products received a lower score than the control, they obtained a more-than-satisfactory rating. The panelists gave similar approval ratings to the tastiness and flavor of all tested cookies, except for the elderberry and hawthorn cookies, which were not as well-liked as other products, including the control. The appearance of the chokeberry, hawthorn, sea buckthorn, and elderberry cookies received lower ratings compared to the control. The color ratings of the enriched cookies, on the other hand, were not significantly different from those of unenriched products. However, this study did not establish the optimal addition of the analyzed by-products to the cookies, and the evaluation was conducted with the same percentage of each fruit powder to the dough.

Other authors [38] conducted a sensory panel evaluation of cookies supplemented with freeze-dried chokeberry using qualitative descriptive analysis. The intensity of the selected attributes was tested on a scale of 1 to 10. Furthermore, a hedonic consumer test was conducted based on a nine-point hedonic scale. Experts positively assessed the quality of the enriched shortbreads with 5%, 10%, and 15% of freeze-dried chokeberry, with scores of 7.4, 8.4, and 6.3, respectively. With increasing additive proportion, the cookies became gradually darker, which was related to the anthocyanin pigments present in the raw material. Moreover, a sweet taste was clearly perceptible at the 5% inclusion level. Increasing the chokeberry powder proportion improved the astringency, chokeberry flavor, and sour flavor intensity. The addition of 5 and 10% fruit resulted in crispness values remaining high, which was similar to the results for the control. Moreover, cookies with 15% of chokeberry powder showed lower crispness, possibly related to the reduced fat content of the final product. The authors found that the sample fortified with 10% powder balanced all the analyzed characteristics with a significant overall quality. Interestingly, the concentration of chokeberry had no significant effect on the overall consumer acceptability of the cookies. The authors’ meticulous examination of the sensory properties of the cookies in this study is noteworthy, but they did not carry out a consumer hedonic test of the control cookies (without the chokeberry powder).

In another study [41], a team of experienced panelists tested the intensity of sensory attributes of Japanese quince-powder-enriched cookies using an unstructured 10 cm linear scale from “no intensity” to “very high intensity.” Additionally, consumer acceptability of the fortified products was assessed on a nine-point hedonic scale. Both analyses were conducted 24 h after baking and after 16 weeks of storage. Sour and lemon flavor intensity ratings improved with increasing lyophilized quince, which is related to the richness of the volatile compounds of the raw material. Citrus and sour flavor intensity scores also increased with the additive, whereas buttery and sweet flavors decreased. The addition of Japanese quince also resulted in a decrease in the cookies’ fracturability, possibly due to the high concentration of fiber, including pectin and cellulose-like polysaccharides. Generally, cookies containing 1 and 1.5% Japanese quince were more acceptable to consumers than control products and those containing 6 and 9% [41]. The storage period generally affected the evaluation of the many sensory attributes’ intensity, mainly reducing their value. The sensory panel identified a decrease in the intensity of citrus, sour, and buttery flavors, as well as buttery and aromatic flavors, although not in all samples. The cookies also became less crumbly, slightly gummy, and softer under storage, mainly those enriched with a high powder concentration (6 and 9%). The authors pointed out this might be linked to the enhancement of hydration properties caused by the presence of fiber, which consequently leads to more interactions with water through hydrogen bonds. However, in the presented study, the authors stored the cookies in cardboard boxes without indicating the relative humidity during storage, which could have caused an increase in their moisture content and consequently changes in their physicochemical properties, negatively affecting the sensory acceptability of the cookies. The sensory evaluation of the cookies containing jujube (*Zizyphus lotus*) confirmed the colorimetric measurements and the instrumental data related to hardness and crispness. Sensory acceptability expressed by color and taste parameters increased by jujube fruit enrichment. On the other hand, texture and aftertaste evaluation were reduced after the powder introduction. Most importantly, the overall acceptability of the cookies did not change significantly after enrichment, except for a reduction in the value of this parameter in products containing 15% of jujube powder [45]. The sensory evaluation (using a nine-point hedonic scale by semi-trained judges) showed that the texture and color parameters of the dragon-fruit-enriched cookies had lower scores if compared to the control [50]. The differences may be linked to the high water absorption and the dark brown color of the powdered dragon fruit. The panelists did not tolerate it. The taste of the cookies improved slightly, due to the mild Kiwi-ish taste. In another study [54], under the influence of papaya powder fortification, the cookies were similarly rated by consumers in crispness and general acceptability when compared to the product without the additive. On the other hand, taste ratings decreased slightly from a score of 8 to 7 (on a nine-point hedonic scale). Furthermore, the overall acceptability of cookies containing date palm fruit powder in the sensory test was not significantly affected by date palm fruit, except for 50% supplementation, compared to the control (refined flour cookie with no additive) or cookies made with whole flour alone [56].

In summary of the above works, it can be concluded that enriching cookies with powdered fruits has a significant effect on their sensory characteristics. Therefore, a nine-point hedonic test is most often applied to evaluate consumer preferences for cookies enriched with these additives, as it allows for more nuanced and differentiated ratings. This test enables participants to express subtle differences in their preferences, providing more precise information for researchers and producers.

### 3.2. Cookies with FBP

Many studies in recent years have also focused on the sensory properties evaluation of cookies enriched with FBP. In the work conducted by Rocha Parra et al. [118], a group of untrained consumers participated in the sensory assessment of the 15% apple pomace cookies, using a nine-point hedonic scale. There were no significant differences in appearance, aroma, texture, or overall appreciation scores. However, the additive effect was positive on taste, regardless of the size of the particles used. Zlatanović et al. [60] assessed the sensory properties of apple pomace powder cookies using a point method on a scale from 0 to 5, where parameters regarding appearance, texture, and taste were analyzed. In general, cookies enriched with coarse flour received higher ratings from consumers than those enriched with fine flour. The differences in particle size distribution and processing of the apple pomace in these two studies may have led to a noticeable difference in the effect of particle size on the sensory evaluations of the cookies. Baked products enriched with 25% apple pomace coarse flour were rated the highest in overall sensory quality [60]. They scored highest in all tested parameters, and their values were greater or not significantly different from the results of the control sample. The cookies with 25% powder additive showed a standard appearance, adequate aroma, and optimal texture. Interestingly, the product supplemented with 50% coarse apple pomace flour also was within “excellent quality”. The other samples and the control were of “very good quality”. The products containing 25 and 50% pomace additives showed the slightest change in sensory properties over the 12-month storage period. Following the sensory quality assessment, cookies containing 50% of coarse powder were finally selected and used for acceptance testing on a nine-point hedonic scale. During the consumer acceptance test, the respondents liked these products, with average hedonic scores above 6 (6.2 ± 1.8). Moreover, the authors identified two significant consumer groups, the first of which felt that the cookies had “too little apple aroma” (41.7% of respondents), while the second admitted that the final product was “not sweet enough” (32.2% of respondents). In another study, ratings for flavor, mouthfeel, texture, and overall acceptability decreased with increasing apple pomace levels [119]. The highest notes for color and flavor were found for 10% powdered cookies, and ultimately, this quantity of additive was deemed optimal. Based on the studies mentioned, it can be concluded that incorporating apple by-products in cookies generally has a positive effect on sensory properties, but this can vary depending on the type and amount of addition. The studies [60,118] revealed that adding 15–50% of apple pomace powder to cookies did not significantly affect aroma, texture, and overall evaluation, and even improved taste. However, another study [119] found that supplementation levels greater than 10% were unacceptable from the consumer’s point of view. The aforementioned differences may be due to variations in apple pomace processing (different temperatures, drying time, and degree of grinding), cookie composition and preparation, heat treatment and exposure time, as well as the individual taste preferences of each group of panelists participating in the study. Therefore, further investigation is needed to determine the optimal amount and form of apple pomace for use in cookies to achieve the best sensory attributes without compromising the texture or other characteristics of the final product.

Another interesting addition to consider is apple peel. In a study by Nakov et al. [62], during the sensory analytical tests (on a five-point scale), the apple peel powder had a positive impact on the aroma and flavor scores of the cookies, while not damaging the smell, appearance, or internal structure ratings. The control sample obtained better grades than the other shortbreads only with regard to texture, except for the cookies with 24% supplementation. The products enriched with 24% apple peel demonstrated the best ratings for flavor, appearance, internal structure, texture, and taste, compared to other supplemented cookies and the control. Substituting wheat flour with 16% and 32% fruit additives also yielded a positive impact on flavor evaluations. In general, it can be inferred that substituting wheat flour with 24% apple peel powder in shortbread pastries may present a trade-off between their chemical and physical properties and sensory evaluation. As the contribution (5, 10, 15, and 20%) of orange peel powder to the cookies increased, the sensory parameter values for color, appearance, aroma, texture, taste, and overall acceptability decreased, but the differences were not significant [72]. Enriched products’ ratings for overall acceptability ranged from 7.30 to 8.30, demonstrating that the semi-educated judges ‘moderately liked’ to ‘very much liked’ the cookies with the additive. The findings suggest that the substitution of wheat flour with powdered orange peel at levels up to 20% can confer functional benefits to cookies while maintaining their sensory attributes. Interestingly, cookies with the addition of 2.5% bergamot pulp powder were most favored by the panelists in overall acceptance when compared to the control and other levels of fortification products [79]. Samples supplemented with 10 and 15% bergamot pulp flour were significantly less acceptable, related to their bitter taste. During the sensory analysis conducted by trained panelists (on a nine-point hedonic scale), the increase in pomace and kinnow peel led to a significant reduction in the flavor and texture notes [120]. Ratings observed for color showed no substantial difference among products enriched with pomace, peels, and the control. The cookies with 10% peel and 5% pomace showed the best overall acceptability, with scores of 8.33 and 8.33, respectively. The outcomes of the aforementioned studies demonstrate that bergamot pulp, as well as kinnow peel and pomace powders, may serve as beneficial supplements for cookies with good sensory acceptance.

Tarasevičienė et al. [65] used a nine-point hedonic scale in the sensory evaluation. The shortbread cookies with the addition of 20% strawberry pomace scored highest in color (7.83 points), aroma (7.83 points), and texture (7.33 points). However, products with 10% raspberry pomace flour were the tastiest (7.33 points). On the contrary, cookies that were enriched with 15% red currant pomace flour received lower ratings in terms of color, aroma, texture, and flavor, as compared to other enriched cookies and the control group. Consequently, it can be deduced that strawberry and raspberry pomace flours would make more suitable functional additives for shortbreads with regard to consumer acceptance, which is a crucial factor that affects purchase decisions. In other studies [29], the extra addition of passion fruit epicarp flour did not significantly modify the flavor ratings of the cookies assigned by the trained panelists on a nine-point hedonic scale. In addition, the flavor, texture, and overall acceptability scores of the products with 3 and 6% admixture were not considerably different from those of the control. Only the supplement at the level of 9% significantly reduced the value of these parameters. The aroma of enriched cookies was judged worse than the control due to astringency. The results of the appearance evaluation of the cookies did not vary significantly between the baked product without the additive and the one containing 3% of epicarp. The 6 and 9% addition of fruit flour caused a reduction in this parameter, which correlates with the markedly darker color of the cookies. Passion fruit epicarp powder can serve as a suitable functional additive for shortbread pastries at levels of 3% and 6%, providing added nutritional and health-promoting benefits while maintaining most sensory attributes at acceptable levels. Based on the results of the preference ranking test in other studies [121], it was determined that there were no discernible differences between the control cookies and those that were fortified with by-products derived from pineapple, apple, and melon [121]. Consequently, cookies that contained the highest concentration of the supplement (15%) were chosen for the acceptance test. The addition of fruit powder did not have any impact on the ratings of the appearance and flavor of the samples. However, pineapple and apple powder improved the flavor of the cookies, whereas the addition of melon rind had an adverse effect, which may have been attributable to the acidity and bitterness of the additive. Ratings for texture and overall acceptability were comparable for all additives, with the exception of melon by-product powder, which had a negative impact on the scores. Moreover, the cookies with a 15% pineapple substitution achieved the highest scores in sensory evaluation among the tested samples. To summarize, based on the findings of this study, it can be concluded that pineapple and apple by-product powders are suitable for fortifying shortbread cookies at a concentration of 15%. However, further research is necessary to determine the optimal amount of melon peel additive to enhance the sensory characteristics of the product.

Sensory evaluation (on a nine-point hedonic scale) revealed that cookies containing 4 and 6% of grape pomace were best accepted by the partially trained panelists (also compared to the control) [96]. The fortified products with 2% powder showed an indifferent taste compared to the control (pomace was not perceptible). In contrast, cookies containing 8% powder demonstrated a bitter taste and worse texture. In other studies [97], the acceptance test on a nine-point hedonic scale was applied. The purchase intention of the cookies manufactured with grape pomace, on the other hand, was determined using a seven-point structured scale, ranging from 7 (I would definitely buy) to 1 (I would definitely not buy). There were positive responses from panelists regarding the enriched products, highlighting the excellent acceptance of cookies containing 19.8% pomace flour from three grape varieties. The authors revealed a high frequency of responses on the “I liked it a lot” scale for odor (31.2%), texture (30.4%), taste (25%), and overall impression (28%). The positive response frequency was highest for the purchase intention attribute and was 55.2%. The most common reply was ‘buy’ (six points on a structured scale) and the negative response rate was very low (15.2%) [97]. Grape seeds could be incorporated into the cookies at a level of 5% without significantly impacting their overall acceptability, based on the sensory evaluation conducted by Kuchtová et al. [23]. In contrast, the ratings for this parameter decreased with increasing grape peel levels. It can be inferred that consumers may generally show interest in purchasing cookies enriched with grape by-products. Moreover, if added at optimal concentrations, pomace and seeds may have a neutral or positive impact on the sensory properties of shortbreads. The aforementioned studies propose that substituting wheat flour with pomace up to a maximum level of 6% and incorporating seeds at a concentration of 5% could yield a final product with a high level of consumer acceptance.

Sensory evaluation of shortbread cookies enriched with sea buckthorn by-product powder was performed using a scale of 1.0 to 10 [122]. Panelists found the cookies containing 20% of the additive to have the most pleasing sea buckthorn and sour flavors. Moreover, the control cookies were rated as the hardest and crumbly. In the sensory evaluation of structural properties, hardness decreased with increasing sea buckthorn biomass content, as confirmed by the results obtained with the texture analyzer. Most importantly, the highest overall acceptability was achieved by cookies that consisted of 15% powder compared to products without the additive. In other studies [101], semi-trained judges, during the organoleptic evaluation (using a nine-point hedonic scale), generally rated the unfortified cookies higher in flavor, aroma, color, and overall acceptability attributes, compared to the chiku pomace products, especially at higher levels of supplementation. The authors observed the opposite effect with the texture of the products, but the differences between all samples were not significant. The results of this study showed that the incorporation of chiku fruit pomace into cookies at concentrations of up to 7% resulted in the same outcomes in terms of taste, aroma, and texture when compared to the control. The cake fortified with 7% powder demonstrated the highest overall acceptability in organoleptic tests, indicating an optimal level of supplementation that can reconcile the health-promoting effects of the additive with the consumer acceptability of the end product.

Based on a preliminary sensory evaluation on a nine-point hedonic scale, the authors of another study [106] found that taste and texture scores decreased with increasing goji pomace content. Incorporation at a level of 10% did not significantly change the acceptability of the appearance of the cookies to consumers, while the others were less liked. The aftertaste and the type of aftertaste were measured on a three-point scale. Cookies with the highest level of incorporation (40%) had the lowest ratings of taste and texture, which were associated with a strong, unpleasant, bitter, and astringent aftertaste, soft texture, and fibrous/grainy taste. In general, cookies can be enriched with goji berry by-product powder up to a concentration of 10% without a significant impact on consumer perception of the products. However, if a higher level of supplementation is desired, further experiments are necessary to improve the organoleptic characteristics of the final cookies. The sensory properties of shortbread pastries were evaluated by other researchers who examined the impact of kiwi powder and its by-products [109]. The 5% and 10% content of both edible portions and kiwi fruit peels did not significantly reduce the notes (on a nine-point hedonic scale) regarding appearance, compared to the control. The cookies with the 15% kiwi fruit edible portion also obtained the highest scores for odor (8.81) among the samples, which is related to the presence of volatile constituents in the fruit. The additions of 5, 10, and 15% of edible portion powder and 5 and 10% of peel caused no significant differences in the flavor acceptability of the cookies. Texture ratings were also similar to those of the control, except for the highest levels of supplementation (20% of edible portion and 15 and 20% of peels). The overall acceptability of cookies containing 5, 10, and 15% edible portion and 5 and 10% of peels was not significantly different compared to the control. Replacing wheat flour with kiwi fruit powder (including the edible part and peel) at levels of 10% and 15% resulted in an improvement in the antioxidant activity and nutritional quality of the cookies while maintaining acceptable sensory properties. The ratings of the cookies enriched with 7.5% cactus pear peel showed no significant difference in comparison to the control sample, as confirmed by the sensory evaluation (on a nine-point hedonic scale) [110]. Ratings of these cookies in appearance, color, taste, flavor, and, most importantly, overall acceptability indicated that the panelists liked them very much, with values of 8.1, 8.15, 8.15, 8.1, and 8.1, respectively. In addition, the researchers demonstrated that a product containing 10% cactus pear peel powder was not acceptable. The aforementioned findings suggest that the incorporation of cactus pear peel powder into cakes at a concentration of 7.5% may strike a balance between the concentration of bioactive compounds and the consumer acceptability of the final product.

For sensory evaluation of cookies enriched with fresh lemon peel, extract from lemon pomace, and a mix of them, a preference test was performed by judges experienced in this type of analysis [74]. The cookies acquired a bitter taste under the influence of the enrichment. In addition, the lemon aroma was more strongly perceptible in the supplemented products, indicating good retention of the aromatic components during baking. The replacement of skimmed milk with lemon pomace extract in the cookies resulted in lower scores for the superficial brown color attribute. It may be related to the less intense Maillard reaction and caramelization caused by the proteins and lactose reduction in the dough. Cookies with added extract were also the crispiest compared to other enriched products, relating to the lowest water activity (0.23) of the mentioned cookies. Incorporating lemon peel and lemon pomace extract into cookie formulations can increase their bioactive compound content without significantly impacting their overall appearance or texture, with the exception of color. This results in acceptable products with improved functional properties. Further research could explore the use of lemon by-products at varying levels of incorporation to determine the optimal concentration.

Considering the reports above, the enrichment of shortbread cookies with powdered and dried fruits and FBP almost always affects their sensory attributes. Fruit additives usually change the flavor and taste, and consequently decides the overall acceptability of cookies.

Table 1 summarizes the effect of the supplementation of shortbread cookies with fruits and FBP on their chemical composition, antioxidant properties, and sensory characteristics of the cookies. In most of the studies cited, the authors determined the optimal addition of fruits to cookies based on sensory evaluation results. However, in studies [21,22,65,74,82], it was not specified, which diminishes the practical value of these works. Furthermore, in the study [61], cookies made with the addition of blueberry powder were not acceptable to consumers. However, the authors only used a single level of addition. Moreover, baking conditions for the cookies are also significant, especially considering the content of bioactive compounds and the antioxidant activity of the obtained cookies. The majority of authors included the temperature and baking time in their studies. However, in studies [61,65,96], the baking conditions were not described. The main benefit of enriching cookies with fruits and FBP, as indicated by the authors mentioned in Table 1, is the increase in phenolic compound content and the accompanying enhanced antioxidant activity, along with an increase in fiber content in the cookies.

## 4. Conclusions

Enriching cookies with fruit additives is a modern dietary approach due to the additives’ high levels of bioactive compounds and nutrients. This strategy may serve as an effective means of preventing diet-related diseases. In particular, the use of FBP is a trend that allows for the utilization of valuable, nutritionally significant substances. Such an approach not only enables better utilization of food resources but also leads to considerably healthier products with improved antioxidant properties. The type of fruit additive and level of substitution can diversely affect the sensory attributes of cookies, especially their color, texture, flavor, and taste, which can impact consumer acceptability. The results of the analysis suggest that various additives, including *Zizyphus lotus* fruit, papaya, date palm fruit, apple pomace, orange peel, passion fruit epicarp, grape pomace and seeds, goji berry pomace, kiwi edible portion and peel, as well as sea buckthorn by-product powders, does not negatively affect consumer acceptance of the final product. It means a promising possibility to use them as functional additives. In recent years, there has been a growing interest in the use of FBP powders in the food industry which, through the processes of drying and crushing, represent a concentrated source of bioactive substances.

## Figures and Tables

**Figure 1 molecules-28-04005-f001:**
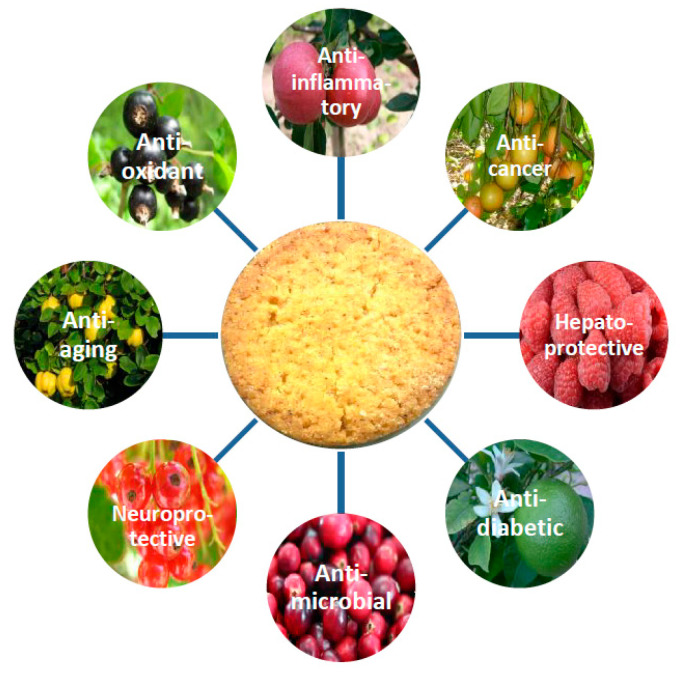
Opportunities to increase the health-promoting benefits of cookies by adding fruits and FBP.

**Figure 2 molecules-28-04005-f002:**
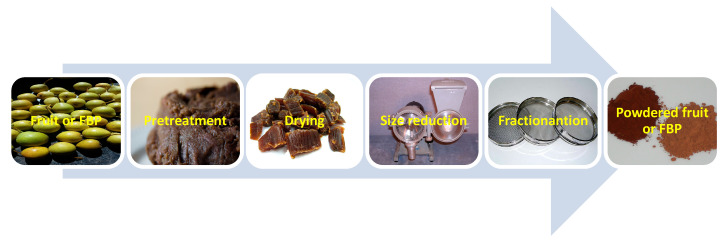
The method of obtaining powdered fruit and FBP.

**Table 1 molecules-28-04005-t001:** Effect of fortifying shortbread cookies with fruit additives.

Main Ingredients and Level of WF Replacement	Cookies Production Method	BTT	Main Enrichment Effect	RAL	Refs.
Wheat flour and camu-camu powder (5, 10, 15, 20%)	The dough was divided into approximately 16 g portions, then formed into circles 6 mm thick and with a diameter of 50 mm.	165 °C, 7 min	TPC and AA increasing.	5–20%	[20]
Wheat flour and *Parinari curatellifolia* peel flour (5, 10, 15, 20%)	Dough with 55–60% moisture content was rolled out to appropriate thickness and transferred to greased baking pan.	150 °C, 20 min	Growth of AA, TFC and TPC. Beneficial effect on thermal properties, crude fiber and ash content. Increase in hardness, decrease in brightness.	-	[22]
Fine wheat flour and grape skin and grape seed powder (5, 10, 15%)	After kneading, the dough was shaped into sheets 2 mm thick, and then cut into pieces (round) 40 mm in diameter.	180 °C, 8 min	Increase in total fiber content (TFC), decrease in hardness.	5%	[23]
Wheat flour and ground chokeberry, hawthorn, sea buckthorn, elderberry, rosehip and rowan (1%).	The dough was stored for 30 min at 4 °C, rolled out to a 3.5 mm thickness and cut into round shapes of the 3.5 cm diameter.	200 °C, 8 min	Increased AA, reduced acrylamide content.	-	[21]
Wheat flour and passion fruit flour (3, 6, 9%)	The dough was prepared using the creaming method, then formed into sheets 3 mm thick and cut into round pieces 45 ± 2 mm in diameter.	170 °C, 15 min	Improvement in AA, TPC and fiber content, with an increase in acrylamide content. Reduction in starch digestion rate (in vitro). Darker color, increase in hardness.	Up to 6%	[29]
Wheat flour and chokeberry powder (5, 10, 15% of the dough)	After dough preparation, replaced 5, 10, 15% of the weight with additive.	180 °C, 10 min	Increase in total polyphenols (TPC), AA, water, ash, reducing sugars. Reduction in fat content.	10%	[38]
Wheat flour and *Zizyphus lotus* powder (15, 30, 45, 100%)	The dough was rolled out to a thickness of 2 mm and sliced into round pieces 5 cm in diameter.	175 °C, 15 min	Improved AA, TPC and TFC.	15%	[45]
Refined wheat flour and pitaya powder (30%, 40%, 50%, and 60%)	After kneading, the dough was left to rest for 10 min, then shaped into balls and cut at roughly 10 g per biscuit.	160 °C, 25 min	Fivefold improvement in fiber content, increase in spread ratio.	50%	[50]
Whole wheat flour and date fruit pulp flour (90:10; 80:20; 70:30; 60:40, 50:50)	Creaming sugar and margarine. The dough was rolled and flattened to a thickness of 3.5 mm, then cut using a hand-cutter.	150 °C, 30 min	Increase in ash, crude fiber and fat content. Decrease in protein content.	30%	[56]
Wheat flour and apple pomace powder (25, 50, 75%)	Wheat flour and apple pomace powder (25, 50, 75%) The dough was cut into cookies 40 mm in diameter and 5–6 mm thick.	175 °C, 10 min	Multi-fold improvements in AA, TPC and TFC.	Up to 50%	[60]
Blueberry powder (37.14% of the dough) and wheat flour	After mixing the components, the dough was rolled to a thickness of 0.5 cm. Next, it was cut into pieces with a diameter of 4 cm.	-	Unmet consumer expectations.	-	[61]
Bread wheat flour and apple peel powder (4, 8, 16, 24, and 32%)	The dough was chilled at 8 °C for 30 min and rolled out to a thickness of 18 mm. Then it was cut into 44 mm diameter cookies.	205 °C, 10 min	Higher AA, TPC, fiber, ash and fat content. Beneficial effect in terms of sensory properties.	24%	[62]
Wheat flour and flour from pomace of strawberries, redcurrants and raspberries (10, 15, 20%)	The dough was rolled out to a thickness of 0.5 cm and cut into rectangles (2 cm wide, 7 cm long).	180 °C, 10 min	Increase in fiber content and hardness.	-	[65]
Wheat flour and blueberry pomace powder (3, 6, 9%)	The dough was cut into round pieces 4 cm in diameter and 0.50, 0.75 and 1.0 cm thick.	160, 170, 180 °C, 12 min	Increase in AA, TPC, fiber content.	9%	[70]
Refined wheat flour and orange peel powder (5, 10, 15 and 20%)	The cake was made using the creaming method.	-	Increase in AA, TPC, total fiber, soluble and insoluble dietary fiber.	Up to 20%	[72]
Wheat flour and fresh lemon peel (10 g, replacing wheat flour) and lemon pomace extract (50 mL, replacing skimmed milk)	The dough was rolled out to a thickness of 3.5 mm and cut into pieces.	180 °C, 7 min	Enhanced AA, TPC and longer induction period.	-	[74]
Wheat flour and *Pastazzo* bergamot pulp flour (2.5, 5, 10, 15%)	Dough was rolled into 3 mm thick pieces with a diameter of 6 cm. Increase in AA, TPC, TFC and water activity. Decrease in pH value. Reduction in baking time.	180 °C, 8 min (FC), 12 min (C)	Increase in AA, TPC, TFC and water activity. Decrease in pH value. Baking time reduction.	2.5%	[79]
Superfine wheat flour and banana peel flour (7.5, 10, 12.5, 15%)	Ingredients mixed.	160 °C, 10–15 min	AA and TPC improvement. Increase in hardness, moisture content and ash. Decrease in the amount of protein and fat. Decrease in brightness and yellowness.	-	[82]
Whole wheat flour and passion fruit peel powder (10, 20, 30%)	Dough was frozen for 1 h, then cut into pieces approximately 50 mm in diameter and 7 mm thick.	180 °C, ±20 min (until fully baked)	Increase medium crude fiber and ash content.	30%	[86]
Refined wheat flour and grape pomace powder (2, 4, 6, 8%)	No information available.	-	Increase in fiber and protein content. Significantly better preservation of texture properties during storage.	4, and 6%	[96]
Wheat flour and grape pomace flour blend (19.8%)	Creaming. Homogenization of ingredients. The dough was opened to a thickness of 6 mm with a dough opener, and then cut into 47.85 mm diameter portions.	150 °C, 10 min	High protein content (75%) and appropriate microbiological quality. Adequate product acceptability and consumer willingness to buy.	19.8%	[97]
Refined wheat flour and chiku pomace (4.5, 7, 9.5, 12%)	After kneading, the dough was rested for 5 min and rolled out to a thickness of 0.44 ± 0.05 cm. Then round 5 cm diameter shapes were formed.	168 °C, 20 min	Increase in AA, crude fiber and dietary fiber. Decrease in hardness and protein content.	7%	[101]
Soft wheat flour folded and ground goji berry pomace (10, 20, 30 and 40%)	Macro wire-cut method. The dough was rolled out and then cut with a cookie cutter.	205 °C, 11 min	Increase in fiber content (soluble and insoluble fraction), free polyphenolic compounds and protein.	10%	[106]
Wheat flour and powder of the edible part and kiwi peels (5, 10, 15, 20%)	Creaming. After mixing the ingredients, the dough was formed into cookies.	170–180 °C, 20 min	Increase in fiber and ash content.	10, and 15%	[109]
Wheat flour (Elsafa) and cactus pear peel powder (2.5, 5, 7.5, 10%)	The dry ingredients were stirred together and blended with fat until achieving a biscuit-like consistency. Subsequently, the egg was folded in, followed by water to create uniform dough that was shaped by cutting.	200 °C, 15 min	Increase in dietary fiber content and antioxidant activity.	7.5%	[110]
Wheat flour and apple pomace powder (15, 30%)	After mixing the ingredients, the dough was left for 30 min. Afterwards, it was laminated and cut into round pieces measuring 40 mm in diameter.	185 °C, 14 min	Reduction in brightness levels, positive impact on taste.	15%	[118]
Straight grade wheat flour and apple pomace powder (5, 10, 15, 20, 25%)	Creaming. The dough was rolled 10 times and then cut into round forms.	185 °C, 20–25 min	Increase in fiber and spread factor.	10%	[119]
Refined wheat flour and pomace powder, kinnow peels (5, 10, 15, 20%) and supercritical liquid kinnow peel extract (1, 2, 3 and 4%)	The dough was cut into round pieces 5 mm in diameter.	190 °C, 15 min	Increase in brightness, crude fiber and ash content. Increase in AA, TPC, TFC and carotenoids. Improved oxidative stability after storage.	10% (PE), 5% (PO), 4% (EX).	[120]
Wheat flour with 10% protein and powder of pineapple central axis, apple endocarp and melon peel (5, 10, 15%)	Dough was rolled out and cut into round pieces.	180 °C, 10 min	Improved nutritional value (fiber content), highest for melon. Decrease in brightness. Positive effect of pineapple powder on acceptance rate and purchase intention.	15%	[121]
Wheat flour and sea buckthorn fruit biomass (pomace) powder (5, 10, 15, 20%)	The dough was rolled out to a thickness of about 3 mm.	180 °C, 8 min	Positive impact on sensory attraction.	15, 20%	[122]
Refined wheat flour and *Haenomeles japonica quince* fruit powder (0.5, 1, 1.5, 3, 6, 9%)	The dough was stored at 4 °C for 24 h. Afterwards, slices 5 mm thick were obtained and round pieces 50 mm wide were cut.	170 °C, 17 min	Increase in AA. Decrease in AA after 16 weeks of storage. Less emission of secondary lipid oxidation products than the control.	1/1.5%	[41]
Refined wheat flour (Maida) and date powder (50, 100%)	After the dough was kneaded, cookies were formed.	160 °C, 30 min	Improve in overall acceptability, quality and physicochemical parameters (moisture, ash, total soluble solids, pH, vitamin C, fat, crude fiber, protein and carbohydrates). Enhancement of overall acceptability.	100%	[123]

RAL—Recommended additive level, BTT—Baking Time and Temperature, FC—Fortified cookies, C—Control, PE—Peel, PO—Pomace, EX—Extract, AA—antioxidant activity, TPC—total content of phenolic compounds, TFC—total flavonoid content.

## Data Availability

Not applicable.

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
