# Peer review of "Enrichment of Cookies with Fruits and Their By-Products: Chemical Composition, Antioxidant Properties, and Sensory Changes"

_molecules, 2023, doi:10.3390/molecules28104005_

Round 1
Reviewer 1 Report
Dear Editor,
In the paper entitled “Enrichment of cookies with fruits and their by-products: chemical composition, antioxidant properties, and sensory changes” the authors raise an interesting issue related to the presentation of the current state of knowledge on the use of various parts of fruit added in various amounts and forms to wheat cakes. The authors analysed the data from many studies clearly pointing to the advantages of including fruits, as they can significantly increase the nutritional value of confectionery products without impairing their sensory characteristics, and even improving them. I believe that before the paper is published in Molecules, the authors should organize the text and give it a clear and transparent narration, which in the current version of the paper is quite difficult to see.
In the introduction to the article, in addition to the positive aspects related to adding fruit preparations to cookies, it is worth describing in a synthetic way the difficulties and limitations associated with this process.
Although the paper is written in good scientific language, it is difficult to find clear connections between the described examples. In section 2, the authors discuss over 9 pages how different amounts and forms of addition of many different fruits affected the levels of phenolic compounds and the antioxidant properties of the cookies. It is a description of observations coming from the articles of other researchers, but it lacks the author's view and finding some similarities and differences, which could lead the authors to their own valuable observations, the discussion of which would enrich the text. This would give it a clear logic, because in the current version of the paper, the order of the discussed content does not seem to matter, because it is primarily a summary of various works on the described topic.
The same happens in section 3, where the authors cite the observations of other researchers on the sensory characteristics of cookies with various additives. Also in this part of the work there is no author's perspective and giving the narration a logical order resulting from the authors' goal of proving or refuting certain tendencies or hypotheses. It would be a good idea for the authors to formulate such general sentences as, for example, the sentence given in lines 673-674: “Grape seeds could be incorporated into the cookies at a level of 5% without significantly impacting their overall acceptability, based on the sensory evaluation conducted by Kuchtova et al. [93].” The authors could make such observations also in section 2, and then cite works confirming or contradicting them, which could lead to further observations and a fluid narrative clearly formulated by the authors throughout the all text. Table 1, prepared by the authors, could be used for this purpose. It should be located at the beginning, not the end of the paper, because the insights derived from it would be a good basis for giving the article a clearly visible author's narration.
Detailed remarks regarding my doubts about specific sentences or words are attached below.
Detailed remarks regarding my doubts about specific sentences or words are included in the manuscript I have attached.
1) line 98 "fruit themselves", it is better to specify that "in the same amount of fruits themselves"
2) line 105-106 "The highest amount of dietary fiber was found in sea buckthorn shortbreads (6.36% 105 dm), while the lowest was in elderberry cookies (1.69% dm)". It is worth specifying whether these differences result from differences in the level of the above-mentioned ingredients in these two raw materials, or from the part of the fruit used and/or the ways of adding them to the dough.
3) line 132 "freeze-dried Japanese quince fruit", the same amount of freeze-dried...
4) line 132-133 "The characteristics of the chemicals created during the Maillard process could contribute to clarifying it". The explanation given below does not apply to the Maillard reaction mentioned in this sentence. Please provide justification related to Maillard reactions.
5) line 143 "and polyphenolic compounds", Tannins aren't polyphenolic compounds?
6) line 184-185 "However, there was no difference in crude fiber concentration between cookies containing 10% of the admixture and between whole wheat flour alone and refined flour (control) products. ". Please indicate what this is due to?
7) line 197 "This section is not mandatory but can be added to the manuscript if the discussion is unusually long or complex.", ?....
8) line 209, line 211 "coarse", coarsely ground pomace flour?
9) line 212 "(fine)", finely ground pomace flour?
10) line 217 "with the food matrix", or oxygen and light
11) line 257 "related", ...depended on the total pholyphenolic content....?
12) line 295 "Flavonoids", Flavonoids are not polyphenols?
13) line 324 "exocarp", fruit peel?
14) line 352 "scores", content?
15) line 400 "Furthermore, pomace powder-containing products", It's about a product with what amount of pomace addition?
16) line 458, line 475 "phenols", it should be "phenolic compounds"
17) lien 474 "phenols", Rather "phenolics" and they are also mentioned along flavonoids and tannins which also are phenolics. By using this word, did you mean the total content of phenolic compounds?
18) line 483 "cactus", cactus or cactus pear?
19) line 489 "cactus", cactus or pear?
20) line 498 "there were no significant differences in DPPH tests", it is worth justifying this in the context of the previous sentence.
21) line 506-507, it is worth justifying it.
22) line 626, The above sentence states that the texture of all the cookies with the additives was inferior, which contradicts this term.
23) Table 1, The table should be placed in the initial part of the paper, because discussing the relationships indicated in it can give the entire article a clearly visible author's narration.
Author Response
Answers for Reviewer 1
Thank you for taking the time to review our work and for all the suggestions that will help us improve it. We hope that our responses satisfy you.
In the paper entitled “Enrichment of cookies with fruits and their by-products: chemical composition, antioxidant properties, and sensory changes” the authors raise an interesting issue related to the presentation of the current state of knowledge on the use of various parts of fruit added in various amounts and forms to wheat cakes. The authors analysed the data from many studies clearly pointing to the advantages of including fruits, as they can significantly increase the nutritional value of confectionery products without impairing their sensory characteristics, and even improving them. I believe that before the paper is published in Molecules, the authors should organize the text and give it a clear and transparent narration, which in the current version of the paper is quite difficult to see.
As per the reviewer's suggestion, appropriate corrections were made, and the text was better organized.
In the introduction to the article, in addition to the positive aspects related to adding fruit preparations to cookies, it is worth describing in a synthetic way the difficulties and limitations associated with this process.
The introduction was corrected.
Although the paper is written in good scientific language, it is difficult to find clear connections between the described examples. In section 2, the authors discuss over 9 pages how different amounts and forms of addition of many different fruits affected the levels of phenolic compounds and the antioxidant properties of the cookies. It is a description of observations coming from the articles of other researchers, but it lacks the author's view and finding some similarities and differences, which could lead the authors to their own valuable observations, the discussion of which would enrich the text. This would give it a clear logic, because in the current version of the paper, the order of the discussed content does not seem to matter, because it is primarily a summary of various works on the described topic.
The same happens in section 3, where the authors cite the observations of other researchers on the sensory characteristics of cookies with various additives. Also in this part of the work there is no author's perspective and giving the narration a logical order resulting from the authors' goal of proving or refuting certain tendencies or hypotheses. It would be a good idea for the authors to formulate such general sentences as, for example, the sentence given in lines 673-674: “Grape seeds could be incorporated into the cookies at a level of 5% without significantly impacting their overall acceptability, based on the sensory evaluation conducted by Kuchtova et al. [93].” The authors could make such observations also in section 2, and then cite works confirming or contradicting them, which could lead to further observations and a fluid narrative clearly formulated by the authors throughout the all text. Table 1, prepared by the authors, could be used for this purpose. It should be located at the beginning, not the end of the paper, because the insights derived from it would be a good basis for giving the article a clearly visible author's narration.
As per the reviewer's suggestion, appropriate corrections were made, and the text was supplemented with our observations and conclusion from the described studies. Moreover, Table 1 was located at the beginning of the manuscript.
Detailed remarks regarding my doubts about specific sentences or words are attached below.
Detailed remarks regarding my doubts about specific sentences or words are included in the manuscript I have attached.
1) line 98 "fruit themselves", it is better to specify that "in the same amount of fruits themselves"
An adequate correction was made. The authors emphasized that the concentration of polyphenols in enriched cookies is lower than the amount of these compounds in the same amount of fruits themselves.
2) line 105-106 "The highest amount of dietary fiber was found in sea buckthorn shortbreads (6.36% 105 dm), while the lowest was in elderberry cookies (1.69% dm)". It is worth specifying whether these differences result from differences in the level of the above-mentioned ingredients in these two raw materials, or from the part of the fruit used and/or the ways of adding them to the dough.
An adequate explanation has been added: The difference was likely due to the varying amounts of dietary fiber present in the different raw materials used. [32,33].
3) line 132 "freeze-dried Japanese quince fruit", the same amount of freeze-dried...
An adequate correction was made: Surprisingly, cookies enriched with 9% of the quince powder had a more powerful radical scavenging effect than the same amount of freeze-dried Japanese quince fruit.
4) line 132-133 "The characteristics of the chemicals created during the Maillard process could contribute to clarifying it". The explanation given below does not apply to the Maillard reaction mentioned in this sentence. Please provide justification related to Maillard reactions.
Thank you for this comment. The adequate explanation was added: The authors explained this phenomenon by the enrichment of cookies with polyphenols, as well as the creation of Maillard reaction compounds. These reactions occur between amino acids and reducing sugars, resulting in the formation of brown pigments and aroma compounds, ultimately leading to an increase in the antioxidant activity of the enriched cookies [42].
5) line 143 "and polyphenolic compounds", Tannins aren't polyphenolic compounds?
An adequate correction was made and the word “tannins” was deleted.
6) line 184-185 "However, there was no difference in crude fiber concentration between cookies containing 10% of the admixture and between whole wheat flour cookies alone and refined flour (control) products. ". Please indicate what this is due to?
The following explanation was added “This is likely due to the high standard deviation values obtained by authors during the de-termination of fiber content, which subsequently affected the results of the analysis of variance”
7) line 197 "This section is not mandatory but can be added to the manuscript if the discussion is unusually long or complex.", ?....
Sorry for this mistake. This sentence was deleted.
8) line 209, line 211 "coarse", coarsely ground pomace flour?
It was corrected.
9) line 212 "(fine)", finely ground pomace flour?
It was corrected.
10) line 217 "with the food matrix", or oxygen and light
The following correction was made: “The greater stability of the active ingredients (even after one year of storage) in the larger particles could be due to less contact of the polyphenolic substances with oxygen and light in food matrix”
11) line 257 "related", ...depended on the total pholyphenolic content....?
The adequate correction was made and the sentence was changed to: The high fiber content present in blueberry pomace is also responsible for its high levels of polyphenolic compounds and antioxidant properties.
12) line 295 "Flavonoids", Flavonoids are not polyphenols?
An adequate correction was made.
13) line 324 "exocarp", fruit peel?
An adequate correction was made.
14) line 352 "scores", content?
An adequate correction was made: “Carbohydrate content was the highest of all macronutrients”
15) line 400 "Furthermore, pomace powder-containing products", It's about a product with what amount of pomace addition?
The adequate correction was made: “Furthermore, 6% pomace powder-containing products had a higher TPC (4.03 mg GAE g−1 dm) than products containing 4% admixture (3.42 mg GAE g−1 dm) and the control sample (3.41 mg GAE g−1 dm)”
16) line 458, line 475 "phenols", it should be "phenolic compounds"
An adequate correction was made.
17) lien 474 "phenols", Rather "phenolics" and they are also mentioned along flavonoids and tannins which also are phenolics. By using this word, did you mean the total content of phenolic compounds?
Yes, w did. An adequate correction was made.
18) line 483 "cactus", cactus or cactus pear?
The authors used two expressions - cactus peel and cactus pear peel – We corrected every expression to “cactus pear powder”
19) line 489 "cactus", cactus or pear?
The authors used two expressions - cactus peel and cactus pear peel – We corrected every expression to “cactus pear powder”
20) line 498 "there were no significant differences in DPPH tests", it is worth justifying this in the context of the previous sentence.
The following explanation was a added. The variations in the outcomes of diverse antioxidant assays can be ascribed to the fact that these assays assess distinct facets of the antioxidant capacity of the compounds under scrutiny. The DPPH assay quantifies the scavenging activity of stable free radicals, while the FRAP method illustrates the antioxidant potential of reducing iron ions, and the ORAC assay, being a competitive technique, measures the oxygen radical absorption capacity [115].
21) line 506-507, it is worth justifying it.
It was justified as fallow: “The observed outcome can be attributed to the potential to the cleavage of lignin-phenolic acid bonds or the degradation of lignin itself at higher drying temperatures, which may lead to the increased release of phenolic compounds from plant tissues. Moreover, the elevated temperature may inactivate the activity of enzymes, increasing phenolic compounds recovery”
22) line 626, The above sentence states that the texture of all the cookies with the additives was inferior, which contradicts this term.
It was improved as fallow: Only regarding texture the control sample obtained better grades than the other shortbreads, except 24% supplementation. The products enriched with 24% apple peel demonstrated the best ratings for flavor, appearance, internal structure, and texture and taste, compared to others supplemented cookies and control.
23) Table 1, The table should be placed in the initial part of the paper, because discussing the relationships indicated in it can give the entire article a clearly visible author's narration.
As per the reviewer's suggestion, the appropriate correction was made, and Table 1 was located at the beginning of the manuscript.
Reviewer 2 Report
The manuscript presents relevant information about fruits by-products in cookies application. However, I suggest some important corrections in order to improve the proposal of this review.
The manuscript not mentioned details about particle size influence in physicochemical properties of cookies, as well as, about preparation of fruit by-products powder. This comment appears in the end of the abstract and there was no much discussion about it.
Author Response
Answers for Reviewer 1
Thank you for taking the time to review our work and for all the suggestions that will help us improve it. We hope that our responses satisfy you.
The manuscript presents relevant information about fruits by-products in cookies application. However, I suggest some important corrections in order to improve the proposal of this review.
The manuscript not mentioned details about particle size influence in physicochemical properties of cookies, as well as, about preparation of fruit by-products powder. This comment appears in the end of the abstract and there was no much discussion about it.
We deleted this sentence from the manuscript. We mentioned in some places these aspects. However, we will not enwiden this topic, because it will be an additional the material for other review paper.
Round 2
Reviewer 1 Report
Dear Editor,
The text of the re-reviewed article entitled “Enrichment of cookies with fruits and their by-products: chemical composition, antioxidant properties, and sensory changes” is now better organized and it is easier to read, although in some places it is too vague and describing various fruit additives could be compared together paying attention to similarities and differences between them. However, I accept its current structure, although I still have doubts about the location of Table 1. In the previous review, I suggested placing it at the beginning of the paper, which the authors did, but they never referred to it in the rest of the text. What was the point of developing it then? I suggest either adding references to this table in the discussion of literature data placed below it, or moving it to the end of the paper with a short commentary on the data presented in it and their connection with the entire text (especially recommended additive level given in the penultimate column of the table).
In the current version of the manuscript, there are also repetitions of the same content and a few stylistic errors, which probably indicates that the authors did not carefully read the entire text of the paper after making corrections. In the attached manuscript I have indicated doubtful fragments that I found but I would like to ask the authors to carefully read the entire paper before sending it to the Editorial Office in its final version.
I think that after applying the indicated corrections, the article will be suitable for publication in the Molecules and will be a valuable contribution to the knowledge on functional food additives.

Author Response
The text of the re-reviewed article entitled “Enrichment of cookies with fruits and their by-products: chemical composition, antioxidant properties, and sensory changes” is now better organized and it is easier to read, although in some places it is too vague and describing various fruit additives could be compared together paying attention to similarities and differences between them. However, I accept its current structure, although I still have doubts about the location of Table 1. In the previous review, I suggested placing it at the beginning of the paper, which the authors did, but they never referred to it in the rest of the text. What was the point of developing it then? I suggest either adding references to this table in the discussion of literature data placed below it, or moving it to the end of the paper with a short commentary on the data presented in it and their connection with the entire text (especially recommended additive level given in the penultimate column of the table).
In the current version of the manuscript, there are also repetitions of the same content and a few stylistic errors, which probably indicates that the authors did not carefully read the entire text of the paper after making corrections. In the attached manuscript I have indicated doubtful fragments that I found but I would like to ask the authors to carefully read the entire paper before sending it to the Editorial Office in its final version.
I think that after applying the indicated corrections, the article will be suitable for publication in the Molecules and will be a valuable contribution to the knowledge on functional food additives.
Dear Reviewer,
Thank you once again for thoroughly reviewing our work. We agree with all the comments, and all suggestions have been taken into account. We apologize for these shortcomings, but we were simultaneously working on another project and had limited time for revisions. Some errors also occurred during the conversion of the Word file to PDF. Below, we have provided detailed comments regarding the changes made.
Thank you for this comment, we decide the move Table 1 to the end of the manuscript and shortly describe it.
Line 113 “in the fruits” was deleted.
Line 114 “described in the literature” was deleted.
Lines 122-123 The sentence “The highest amount of dietary fiber was found in sea buckthorn shortbreads (6.36% dm), while the lowest was in elderberry cookies (1.69% dm)” was deleted.
Lines 156-157. The sentence “This also supports the incorporation of FCP into cookies”, was changed to: “There are additional factors that support the use of aronia pulp for enriching cookies”
Lines 206-207: The sentence: “This may also be attributed to the fact that the authors only determined the carbohydrate content of the cookies and did not measure the fiber content”
Lines 267-268: We kept the name of this chapter as it was. We are unsure why the italic font appeared in the PDF version and why the chapter name was incorrectly positioned. In the sent "Word version," such a situation did not occur. It is possible that there was a system error during the creation of the PDF file.
Line 257: “ground apple pomace flour” was deleted
Line 465: “Particularly, the study” was changed to “Particularly, the study demonstrated.
Line 560: The same situation as mentioned previously in the Word file everything is fine. Consequently, we did not include the additional space, because in the world file, everything is OK.
Line 590: “a good source of” was changed to “a rich source of”
Lines 653-659: The same situation as previously: In our world file everything is OK. There is no bolded text and chapter 3 is given in the proper place.
Lines 694-696: The sentence: “Experts positively assessed the quality of the enriched shortbreads, with scores of 8.3 for the 10% sample, 7.4 for the 5% sample, and 6.3 for the 15% sample was changed to”
Lines 701-702: The sentence: “Moreover, 15% of chokeberry cookies showed.... was changed to ” “Moreover, cookies with 15% of chokeberry cookies powder showed”
Line 751-755: The same situation as previously: In our world file everything is OK. There is no italic text and chapter 3.2 is given in the proper place.
Line 827: The sentence: “given that the optimal incorporation level is determined” was deleted.
Line 845: “3% of epicarp substitution” The sentence was corrected to “3% of epicarp . The 6 and 9% addition of”
Line 937: “all” was deleted.
Also after text reading, we corrected a few minor mistakes.